# Adiponectin: The Potential Regulator and Therapeutic Target of Obesity and Alzheimer’s Disease

**DOI:** 10.3390/ijms21176419

**Published:** 2020-09-03

**Authors:** Jong Youl Kim, Sumit Barua, Ye Jun Jeong, Jong Eun Lee

**Affiliations:** 1Department of Anatomy, Yonsei University College of Medicine, Seoul 120-752, Korea; jongyoul74@gmail.com (J.Y.K.); drsbarua@gmail.com (S.B.); jyj4453@naver.com (Y.J.J.); 2BK21 Plus Project for Medical Sciences, and Brain Research Institute, Yonsei University College of Medicine, Seoul 120-752, Korea

**Keywords:** Alzheimer’s disease, metabolic disease, adiponectin, insulin, antioxidants

## Abstract

Animal and human mechanistic studies have consistently shown an association between obesity and Alzheimer’s disease (AD). AD, a degenerative brain disease, is the most common cause of dementia and is characterized by the presence of extracellular amyloid beta (Aβ) plaques and intracellular neurofibrillary tangles disposition. Some studies have recently demonstrated that Aβ and tau cannot fully explain the pathophysiological development of AD and that metabolic disease factors, such as insulin, adiponectin, and antioxidants, are important for the sporadic onset of nongenetic AD. Obesity prevention and treatment can be an efficacious and safe approach to AD prevention. Adiponectin is a benign adipokine that sensitizes the insulin receptor signaling pathway and suppresses inflammation. It has been shown to be inversely correlated with adipose tissue dysfunction and may enhance the risk of AD because a range of neuroprotection adiponectin mechanisms is related to AD pathology alleviation. In this study, we summarize the recent progress that addresses the beneficial effects and potential mechanisms of adiponectin in AD. Furthermore, we review recent studies on the diverse medications of adiponectin that could possibly be related to AD treatment, with a focus on their association with adiponectin. A better understanding of the neuroprotection roles of adiponectin will help clarify the precise underlying mechanism of AD development and progression.

## 1. Introduction

The global prevalence of obesity has increased at an alarming rate over the years. In 2016, it was estimated that nearly 39% of adults aged ≥18 years were overweight worldwide, and 13% were obese [WHO, 2016]. Obesity is generally defined as a body mass index of >30 kg/m^2^ and is mainly caused by physical inactivity and westernized dietary habits. Obesity is a major concern because it is a risk factor for a plethora of metabolic diseases that increase mortality rates. Since obesity causes insulin resistance, it is one of the major risk factors for type 2 diabetes (T2DM). After years of obesity-associated hyperinsulinemia, the insulin secretory function in the pancreas could falter and eventually lead to hyperglycemia. A previous study has suggested that obesity and diabetes are linked to Alzheimer’s disease (AD) [1]. The pooled effect size for AD in relation to obesity and diabetes was calculated at 1.59 and 1.54 in longitudinal epidemiological studies of body mass, metabolic syndrome (dyslipidemia, hypertension, abdominal obesity, and insulin resistance), diabetes, and glucose and insulin levels [2]. A clinical study has also shown that an increased number of metabolic and vascular risk factors in midlife is critical for amyloid deposition and could lead to the risk of developing AD during old age [3].

Recently, the World Alzheimer Report estimated that the total number of people with AD and dementia is set to triple to 132–152 million cases worldwide by 2050 (World Alzheimer Report 2015). About 60%–70% of dementia cases are caused by AD, a chronic neurodegenerative disease. These dementia cases are often associated with extracellular amyloid beta (Aβ) plaques and intraneuronal deposits of neurofibrillary tangles (NFTs) in the brain. Hyperphosphorylation of tau protein leads to the formation of NFTs, whereas the accumulation of Aβ forms hard, insoluble plaques (Aβ peptide) [4]. For over two decades, these two proteins have been the main target for AD therapeutics; currently, available treatments for AD are symptomatic and do not decelerate or prevent the progression of the disease. However, these therapies demonstrate modest but particularly consistent benefits for cognition, global status, and functional ability. The abovementioned studies have suggested that another approach should be considered for the development of AD treatment based on the relationship between AD, obesity, and T2DM. According to the Mayo Clinic AD Patient Registry, 80% of AD patients had either diabetes or showed impairment in glucose tolerance [5]. Many studies are being conducted to decipher the underlying mechanisms responsible for the association between obesity, T2DM, and AD. In some studies that used a rodent model, a high-fat diet was shown to cause AD pathology due to the accumulation of Aβ peptides and phosphorylated tau proteins, as well as cognitive impairment [6,7].

Adiponectin secreted from the adipose tissue sensitizes the insulin receptor signaling pathway and prevents inflammation. Adiponectin is a protein that modulates a number of metabolic diseases (Figure 1), including diabetes, dyslipidemia, atherosclerosis, and comorbid metabolic dysfunction that occur in cardiovascular diseases such as hypertension. Many studies have indicated that alteration levels of adiponectin in the plasma and cerebrospinal fluid (CSF) correspond to a distinctive condition of mild cognitive impairment (MCI) and AD [8,9,10,11,12,13]. The reason for the discrepancies among these studies is unclear, but adiponectin may be considered to be a metabolic biomarker for AD.

In this review, we will discuss the potential mechanisms that bridge the relationship between adiponectin and AD. Understanding these mechanisms could narrow down the process of AD’s therapeutic window and also help to design novel therapeutic applications against AD and related neurodegenerative diseases.

## 2. Relationship between Metabolic Disorder and Alzheimer’s Disease

The most common age-related neurodegenerative disease is AD, which accounts for the most predominant form (about 60–70%) of dementia, as mentioned by WHO. It is characterized by a gradual loss of learning and memory, particularly episodic memory, and may lead to death within 10 years of onset [14,15]. Limited information is known about the etiology and associated nongenetic risk factors of AD. However, some modifiable events, such as dietary habits, lifestyle, and environmental exposure (toxic chemicals), and the nonmodifiable event of aging are common global factors for AD pathology [16,17]. The prevalence of AD is higher in developed countries than in developing countries, and >40% of AD cases are known to occur in individuals aged ≥80 years.

AD was named after Alois Alzheimer, the first person to describe that extensive neuronal tangles and amyloid plaques, considered to be important hallmarks of the disease, can be found in the AD brain. However, recent studies have shown that AD patients have an insidious loss of neurons and increased reactive gliosis in different parts of the brain, comprising the cerebral cortex and the limbic system, including the amygdala and hippocampus [18]. Cellular changes in the basal ganglia, cerebellum, brain stem, and spinal cord are relatively spared in AD. According to pathological findings of affected AD brain regions, two different types of aggregates are commonly found in the intracellular and extracellular compartments—the intracellular aggregates are known as NFTs and the extracellular aggregates are known as senile plaques that consist of insoluble paired helical filaments of hyperphosphorylated tau protein [19,20]. The major component of the senile plaque is known as Aβ aggregation, which occurs before other pathological events such as NFT formation and neuronal death [21]. Thus, Aβ aggregation is considered the central player of neuronal death, which has also been observed in in-vitro neuronal cells subjected to synthetic Aβ peptide (Aβ1-42/43) treatment [22]. Over the years, the amyloid hypothesis, a process where extracellular Aβ accumulation is followed by the loss of synapses and subsequent neuronal loss, has become the popular concept of AD pathogenesis and has been used by many AD research groups [23].

It has been suggested that metabolic impairment and modification of the AD-related protein levels can contribute to a higher prevalence of AD. Dyslipidemia, hypertension, abdominal obesity, and insulin resistance are the popular hallmarks of metabolic abnormalities, which are collectively known as metabolic syndrome (MetS) [24]. Individuals with higher life expectancy (above 80 years) have a higher prevalence of MetS, which is a potential cause of neurodegenerative diseases, especially cognitive impairment and dementia. Other studies have also found that MetS is not solely responsible for AD progression in elderly people, regardless of their backgrounds. Higher MetS could bridge inappropriate regulation of the central nervous system (CNS) function (especially part of the limbic system, such as the prefrontal cortex, hippocampus, and amygdala) and metabolic regulatory responses that result in functional cognitive impairments [25]. Metabolic products that are produced in peripheral organs, such as estrogen, insulin, cortisol, and leptin, can cross the blood–brain barrier (BBB) and influence cognitive function. However, neuropeptides such as orexin (mice), allastostatin (drosophila), and neuropeptide Y (grass carp) modulate metabolic processes in different animals [26]. In a recent review, Yi et al. suggested that neuropeptide Y receptors in humans can be a promising target for metabolic disorders [27]. Furthermore, food intake and weight gain can be increased by lesions in oxytocin-containing hypothalamic nuclei [28].

Some studies on metabolic diseases have reported that metabolic diseases, such as T2DM and obesity, are associated with AD. Metabolic alteration of substances such as T2DM-related insulin receptors [29] and obesity-related adiponectin [30,31] could alter the process of aging and age-related dementia, such as in AD. In a recent study, Nasoohi et al. suggested the relationship of “type 3 diabetes” to AD pathology, in which the metabolic syndrome consisting of oxidative stress and neuroinflammation leads to brain insulin resistance [32]. This concept suggests that the thioredoxin-interacting protein (TXNIP) is a key regulator of oxidative stress and inflammasome activation, which is associated with impaired insulin function in the brain. Brain insulin resistance is found to be related to the progressive atrophy of the brain regions of early AD progression, which are cingulate cortices, medial temporal lobe, prefrontal gyri, and other regions [33]. Along with the discussion above, brain insulin resistance due to metabolic disorders such as diabetes mellitus, metabolic syndrome, and nonalcoholic fatty liver disease are considered to be a prominent component of AD pathology; the maintenance of brain insulin supply should also be a target of AD therapy for individuals with metabolic disorder [34]. Thus, metabolism and neurological disorders are closely related to each other.

## 3. Adiponectin as a Modulator of Metabolic Disorder

Adiponectin is secreted with the bioactive molecule leptin from normal adipose tissues (together, they are called adipokines) [35]. It regulates glucose and fatty acid metabolism by increasing insulin sensitivity of peripheral organs [36,37,38]. However, small amounts of adiponectin, an adipose-tissue-specific protein, can also be synthesized by other cell types. Paradoxically, the adiponectin level decreases with the increase of central adiposity, which causes obesity [39,40,41]. Obesity causes damage to several organ systems because it regulates MetS, which can be characterized physiologically as excess weight and pathologically as high triglyceride levels and insulin resistance [42]. Moreover, obesity is related to an increase in cognitive decline and AD [43]. Recently, our laboratory has shown that touchscreen-based behavioral testing of high-fat diet mice led to an impairment in cognitive function compared with cognitive function in normal diet mice [44]. MetS, including obesity, cardiovascular disease, type 2 diabetes, and neurodegenerative disorders, is associated with the regulation of adiponectin expression [8,45,46]. Like other hormonal proteins, adiponectin also functions through specific receptors known as adiponectin receptors. Adiponectin receptors have been classified into adiponectin receptor (AdipoR) 1, AdipoR2, and T-cadherin. Through the recruitment of adaptor protein APPL1 by AdipoRs activation, adiponectin signaling regulates a series of signaling pathways [47]. AMP-activated protein kinase (AMPK), peroxisome proliferator-activated receptor-α (PPARα), IkB kinase (IKK)/NF-κB/PTEN, IRS1/2–Akt, and Ras-ERK1/2 signaling are examples of downstream adiponectin signaling [48,49,50,51,52]. Adiponectin receptors are found to be expressed throughout the whole brain. Thundyil et al. suggested that in the cortical neurons, AdipoR1 expression is more prominent than AdipoR2 expression [53]. On the other hand, T-cadherin acts as the coreceptor of a unidentified receptor, through which the adiponectin can transmit the metabolic signals [54].

## 4. Adiponectin’s Potential Role in Alzheimer’s Disease

### 4.1. Adiponectin and Brain

Adiponectin gives a beneficial effect on synaptic regulation and memory in AD (Figure 2). It also promotes synaptic plasticity in AD by improving the hippocampal’s long-term potentiation [55]. Adiponectin and its homolog, osmotin, enhance neurite outgrowth and synaptic complexity and improve learning and memory defects in mouse AD models [56,57,58,59]. Furthermore, chronic adiponectin deficiency in aged mice leads to AD-like cognitive impairments and pathologies [60]. In a recent clinical study, individuals with higher-than-normal adiponectin levels performed better in cognitive tests, indicating the protective effect of adiponectin against cognitive failure. Furthermore, the study showed that adiponectin could be used to identify the risk of cognitive dysfunction [61]. The upregulation of serum adiponectin expression has been found to be associated with MCI and AD [8]. The adiponectin receptors AdipoR1 and AdipoR2 have approximately 95% homology between human and mice. They are ubiquitously expressed and structurally related in humans and mice, with variable affinity to different isoforms and predominance in some tissues [62]. The expression of AdipoR1 and AdipoR2 is mainly localized to neurons in the hypothalamus, brainstem, and cortex [22], as well as the nucleus basalis of Meynert and the hippocampus, the two main targeted structures in AD [63]. In the hypothalamus and the brainstem, adiponectin is thought to regulate food intake and energy expenditure via AdipoR1-mediated AMPK signaling [23]. However, low levels of adiponectin in CSF may be compensated by the presence of two high-affinity receptors, AdipoR1 and AdipoR2, in the brain [64,65]. Suppression of AdipoR1 can result in metabolic diseases such as obesity and diabetes, which also potentiate spatial learning deficit, memory impairment, and AD pathologies [30]. Hence, studies have evaluated adiponectin and its receptors as therapeutic alternatives for AD.

In AD, Aβ has to cross BBB to be transported in the brain, where it is regulated by specific receptors and transporters [66]. Therefore, it is necessary to protect BBB disruption. Adiponectin protects BBB disruption by inhibiting apoptosis of endothelial cells, protecting tight junction integrity via the AdipoR1-mediated NF-κB pathway, and maintaining the balance of Aβ transporters in endothelial cells [67].

### 4.2. Adiponectin Improves Insulin Signaling

Recently, many studies have provided evidence that insulin signaling dysfunction plays a key role in cognitive decline, such as in MCI and AD [68,69,70]. It is well known that T2DM is independently associated with cognitive dysfunction and loss of hippocampus volume [71]. Further, insulin signaling prevents Aβ oligomer toxicity [72]. Adiponectin has been found to be beneficial for T2DM because of its ability to enhance insulin sensitivity, and it has been used in T2DM treatment [73]. A few studies have reported the relationship between increased diabetes prevalence and decreased levels of adiponectin [74]. In T2DM patients with low adiponectin, the hippocampus volume is significantly decreased [75]. Lower levels of adiponectin in T2DM have also been associated with lower gray matter volume and reduced cerebral glucose metabolism in the temporal brain regions [76]. In addition, adiponectin-deficient mice have been used as models of insulin resistance and the associated memory pathology [60]. In a rat cognitive-deficient model induced by streptozocin, which is commonly used to induce diabetes, adiponectin attenuated tau hyperphosphorylation and alleviated cognitive function by activating the PI3K/Akt/GSK-3β signaling pathway [77].

### 4.3. Adiponectin Regulates Glucose/Fatty Acid Metabolism

Deterioration of cerebral glucose metabolism is an important feature in age-related AD and is key to the progression of AD pathogenesis [78,79,80,81]. Adiponectin modulates glucose metabolism in hippocampal neurons by increasing glucose uptake, glycolysis, and adenosine triphosphate production rates [82]. Glucose and lactate are considered major energy sources in the brain. However, the amounts of glucose consumption and oxygen utilization in the brain are not the same [83]. In addition, lactate cannot generate energy because of the fast removal of lactate from cells and activated tissues [84]. A study conducted by Dhopeshwarka et al. suggested that there are gaps between glucose consumption and oxygen utilization by the brain and that fatty acids can enter the brain and mitochondria and can be oxidized to produce energy [85]. About 20% (maximum) of the total energy in the brain can be produced from mitochondrial oxidation of fatty acids [86]. Moreover, fatty acids are regarded as key players in the homeostasis of glucose [87]. Fatty acid metabolism has been found to be related to MCI and to adiponectin and its receptors in mice fed a high-fat diet [44,88,89]. Therefore, changes in adiponectin levels can alter the brain metabolism and progression of AD. Adiponectin also activates the AMPK and PPARα pathways through AdipoR1 and AdipoR2, respectively, which reduce hepatic lipogenesis and enhance β-oxidation [90].

### 4.4. Adiponectin Alleviates Inflammation

One of the key factors for cognitive decline (MCI) and AD is chronic neuroinflammation [91,92,93]. Amongst renowned anti-inflammatory molecules, adiponectin is considered to be an active contributor to chronic inflammation in obesity and T2DM [91,92,94,95]. Chronic inflammation, which induces AD and metabolic-distress-related pathologies such as neuronal insulin resistance, endoplasmic reticulum stress, synaptotoxicity, and neurodegeneration, is caused by the secretion of proinflammatory cytokines by microglial cell activation [92,96]. In an adiponectin-knockout mouse model, activation of proinflammatory cytokines such as interleukin (IL)-1β, IL-6, and tumor necrotic factor-α has been shown to cause the development of AD-like pathology [60]. Furthermore, adiponectin prevents neuroinflammation by decreasing microglia and regulating the brain macrophage proinflammatory phenotype [97,98]. Therefore, changes in adiponectin levels may be closely related to neuroinflammation in AD.

### 4.5. Adiponectin Has Protective Effect on Oxidative Stress/Hypoxia

Studies have shown that oxidative stress and hypoxia conditions render an important role in the pathogenesis of age-related neurodegenerative diseases such as AD [78,99,100]. Adiponectin alleviates oxidative stress and oxidative-stress-mediated cytotoxicity [101,102] and has a protective effect in high glucose concentrations in blood [26]. Many of these effects have been reported to occur because of upregulated AMPK signaling [101,103]. Since AMPK is considered the general energy sensor in the brain, inhibition of adiponectin may influence the AMPK pathway, which in turn could affect brain metabolism [104,105]. Similarly, in a hypoxic environment in obese individuals, hypertrophic adipocytes upregulate the expression of hypoxia-inducible factor-1α (HIF-1α) [106]. Upregulation of HIF-1α has been found to inhibit the production of adiponectin. This phenomenon has been confirmed by the expression of high levels of adiponectin mRNA in adipocyte-specific HIF-1α-deficient mice fed a high-fat diet for 7 weeks compared with control mice [107]. With the above study, adiponectin can be considered a modulator of neurocognitive disorders, which might suggest adiponectin as a potential therapeutic target for AD.

### 4.6. Adiponectin and Neuroprotection/Neurogenesis

Adiponectin has a neuroprotective effect in various conditions. It shows a protective effect in brain injury caused by ischemic stroke and intracerebral hemorrhage [108,109]. Neuroprotective effects of adiponectin have also been demonstrated in a kainite-induced excitotoxicity model [103]. Furthermore, adiponectin plays a role in many deleterious conditions such as Aβ deposition/tau phosphorylation, neuroinflammation, and oxidative stress by protecting neurons and glial cells. Hippocampal neurogenesis is crucial for maintaining cognitive function; however, it is impaired in AD patients [110,111]. In this respect, it seems necessary to pay attention to the neuroproliferative effect of adiponectin in the adult brain. Intracerebroventricular injection of adiponectin has shown neurogenic and proliferative effects in an adiponectin-deficient mice model [55]. An in-vitro and in-vivo study has also indicated that adiponectin stimulates neurogenesis through AdipoR1 [112].

## 5. Adiponectin-Associated Therapeutic Strategy against AD Induced by Metabolic Diseases

A number of in-vitro, animal, and clinical studies have been conducted to find molecular targets that prevent protein aggregation, oxidative stress, and inflammation for AD treatment. However, a therapeutic target is still unclear. Adiponectin can be considered a protein of interest in the search for new neuroprotective targets for AD. Previous studies have attempted to use adiponectin levels as an AD marker [8,9,10,13,113] (Table). In addition, numerous therapeutic agents that are being considered new paradigms in AD therapy have been found to be related to adiponectin signaling. These therapeutic agents do not only include adiponectin and AdipoR homologs but also conventional AD drugs, anti-insulin resistance drugs, and cardiovascular drugs.

### 5.1. Adiponectin as an AD Marker

In some recent studies, the relationship between plasma and CSF adiponectin levels in MCI or AD has been reported. However, there are discrepancies in their results. Some studies have shown decreased adiponectin levels in AD or MCI [113,114,115], while others have shown increased levels or insignificant changes [9,11,12,13,116,117]. The reason for the discrepancies among these studies might be because of the ambiguous criteria for classifying AD and MCI patients or failure to exclude other factors that may have affected adiponectin levels. As mentioned earlier, conventional AD medications (acetylcholinesterase inhibitors) can increase serum adiponectin levels. Furthermore, there is also a possibility that increased adiponectin levels may have served as a compensatory mechanism for the progress of AD. Thus, to establish the relationship between adiponectin levels and AD, more controlled studies should be conducted.

### 5.2. Adiponectin and Adiponectin Receptor Homolog

Osmotin, a protein found in tobacco that structurally and functionally mimics adiponectin, positively modulates the AdipoR1/AMPK/SIRT1 pathway and reduces the AD-related protein Aβ expression (Figure 3) [59]. SIRT1 and AMPK are known for their metabolic activities and cellular energy homeostasis; they positively regulate each [118,119]. Osmotin treatment has been found to inhibit the expression of AD markers, such as amyloid precursor protein, p-tau, and Aβ, in inflammation-induced mouse brains [82,120]. In addition, adiponectin also enhances neurite outgrowth and synaptic complexity via AdipoR1/NgR1 signaling [57].

Adiporon, an agonist of adiponectin receptors that bind AdipoRs, has been known to play a vital role in many neurological diseases. Its ability as an anti-depressive agent and metabolic regulator has been demonstrated in a mouse model of depression, where it also regulated dopaminergic neurons [121]. It has also been shown to modulate fear and intrinsic excitability in the hippocampus [122]. These beneficial neurological effects are possible because of the BBB-penetrating property of this molecule. More recently, Liu et al. have reported that adiporon improves cognitive dysfunction, inhibits Aβ deposition, and restores impaired hippocampal neuron proliferation activation in AD mice by activating the AdipoR1/AMPK pathway [123]. Ultimately, osmotin and adiporon can be effective and realistic therapeutic alternatives for adiponectin-based AD treatment in patients.

### 5.3. Adiponectin and Conventional AD Drug

The US Food and Drug Administration (FDA) has approved only two types of medications for the treatment of cognitive dysfunction of AD—acetylcholinesterase inhibitors (AChEI; donepezil and galantamine) and memantine [124]. One study has shown that donepezil increases serum adiponectin levels in AD patients, while another study has shown that galantamine increases serum adiponectin levels in a type 2 diabetes rat model [125]. A recent study has also reported a significant weight loss after AChEI treatment and that AChEI treatment exerts an insulin-sensitizing effect via the activation of IR/PI3K/Akt/GLUT2,4 and Wnt/GSK3β/β-catenin signaling [126]. Although more studies are needed, the beneficial effect of AChEI on AD is thought to be possibly mediated by adiponectin.

Memantine is another FDA-approved medication for AD. Although it cannot be factually claimed that its therapeutic mechanism is directly associated with adiponectin, there are some interesting points that suggest a connection between them. Memantine is an *N*-methyl-d-aspartate (NMDA) receptor antagonist that modulates glutamatergic dysfunction and inhibits excitotoxicity mediated by NMDA [127]. Its therapeutic ability in AD is attributed to this mechanism. Similarly, adiponectin has been shown to have a neuroprotective effect by inhibiting NMDA-mediated excitotoxicity [128]. In addition, adiponectin and memantine play a protective role in glutamate-induced excitotoxicity (shown in both animal/in-vitro models) [129,130]. Furthermore, memantine also has the ability to attenuate insulin resistance and improve brain function in high-fat diet-induced models. Given these similarities, there is a possibility that the comprehensive role of adiponectin in AD could be a hint in conducting research on memantine.

### 5.4. Adiponectin and Type 2 Diabetes Medications

Several studies have suggested that AD and T2DM are related to each other; thus, T2DM drugs and their potential to be used as AD treatment are drawing attention [131,132]. There are several types of type 2 diabetes drugs, such as biguanide derivatives (metformin), PPARγ agonists (thiazolidinedione [TZD] derivatives), glucagon-like peptide-1 receptor (GLP-1R) agonists, dipeptidyl peptidases 4 (DPP-4) inhibitors, sodium-glucose transport protein 2 inhibitors, and second-generation sulfonylureas [133]. In addition to these drugs’ ability to improve cognitive function, some of these drugs are capable of increasing adiponectin levels.

Metformin, the drug used as first-line pharmacologic therapy in T2DM, has been shown to attenuate AD-like pathology in both in-vitro and in-vivo studies [134,135,136]. An improvement in cognitive function has been observed in many studies on mild cognitive function and dementia patients [60,137,138]. Many studies have shown that metformin increases serum adiponectin levels in various conditions [139,140,141]. Furthermore, like adiponectin, metformin plays a neuroprotective role by activating AMPK signaling [142,143].

TZD derivatives, including rosiglitazone and pioglitazone, act as PPARγ agonists and are used for T2DM treatment because of their insulin sensitization activity [144,145]. They have also been researched as alternative AD treatments [146,147]. Increased adiponectin by TZD might play a key role in the insulin sensitization activity induced by TZD [132,148]. Thus, TZD’s beneficial effect on AD might be related to the increased adiponectin levels.

An insulin-releasing hormone in hyperglycemic conditions, GLP-1, has been found to have a neuroprotective effect [149,150,151]. GLP-1 receptors are responsible for controlling food intake and body weight and are widely distributed in the brain and special neurons (pyramidal neurons of the hippocampus and neocortex), which means that it plays an important role in neural function and synaptic transmission [149,151,152,153,154]. In an AD mice model, Val(8)GLP-1, liraglutide, and exidin-4 (GLP-1 analogs, well known for the upregulation of adiponectin) treatments rescued synaptic plasticity by preventing synaptic degradation, which is also correlated with the increased learning ability of new spatial tasks [155,156,157,158]. Several studies have also shown the beneficiary effect of GLP-1 analogs for AD [159,160,161].

Inhibitors of DPP-4 also increase GLP-1 signaling activation by inhibiting the degradation of GLP-1 [162]. Therefore, they potentially act on AD, like other GLP-1 analogs [163]. They also lead to increased adiponectin and are thought to be alternative medications for AD [164,165,166,167]. Accordingly, the GLP-1 functional facilitator, GLP-1 analogs, and DPP-4 inhibitors might be good candidates for AD treatment because of their ability to increase adiponectin.

### 5.5. Drugs for Cardiovascular Disease and Adiponectin

A recent study has suggested that medications aimed at cardiovascular diseases may also decrease the risk of dementia caused by AD [168]. Several neuroprotective pleiotropic agents have been shown to increase plasma adiponectin levels [169]. In studies with animal models, angiotensin II receptor blockers have shown beneficial effects on cognitive function related to AD [170,171,172]. Improvements in AD progress and cognitive function have also been observed in human clinical trials [173,174,175]. Moreover, angiotensin II receptor blockers can reduce neuroinflammation directly or by regulating the infiltration of inflammatory cytokines because of their ability to restore the blood-brain barrier [176,177,178]. Altogether, angiotensin II receptor blockers may be beneficial in AD therapy because it upregulates adiponectin [179,180,181].

Angiotensin-converting enzyme inhibitors (ACEIs) regulate the renin–angiotensin system and angiotensin II receptor blockers. Upregulation of blood adiponectin has been observed after treatment with ACEIs [182,183]. Some clinical and meta-analysis studies have also shown that ACEIs reduce the progression and risk of AD [184,185,186,187]. This result has also been observed in several animal model studies [188,189,190]. However, in one meta-analysis study, ACEIs had no effect on cognitive decline [191], and some studies have shown a negative effect of ACEIs in in-vivo AD models [192,193,194]. Thus, ACEI treatment is controversial for its effect on AD.

Another cardiovascular medication, fibrate, a PPARα agonist, has been shown to increase adiponectin levels [195,196]. Since ligand-activated PPARα decreases tau phosphorylation, Aβ pathology, and neuroinflammation, fibrate might be a good candidate for AD treatment [197].

Statins (such as simvastatin, paravastatin, and atorvastatin) have been reported to increase blood adiponectin levels [198,199,200]. Statins are well known for their anti-inflammatory properties in diabetic patients [201,202]. According to the above studies, statins might also be useful in the treatment of AD by increasing blood adiponectin levels and reducing neuroinflammation.

## 6. Conclusions

The conventional approach to developing therapeutics for AD has been focused on the Aβ hypothesis, which insists that Aβ causes AD pathologies. However, this approach has yet to provide a successful treatment or prevention method for AD. Based on the metabolic alterations that occur in the pathological markers of AD, this review suggests that the adipocyte metabolite adiponectin can be considered a therapeutic option for the treatment of AD.

## Figures and Tables

**Figure 1 ijms-21-06419-f001:**
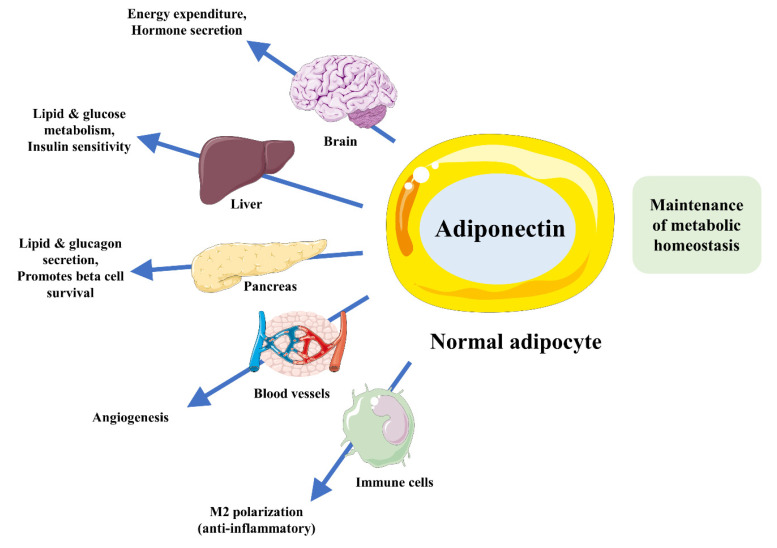
Major mechanisms of adiponectin’s actions in the maintenance of metabolic homeostasis.

**Figure 2 ijms-21-06419-f002:**
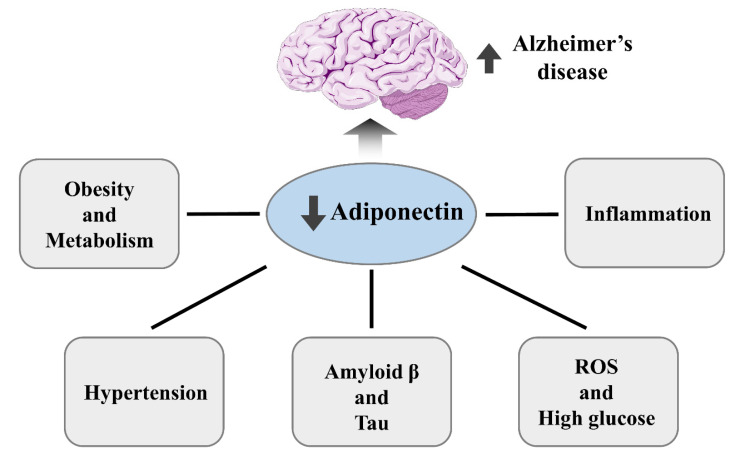
Downregulation of adiponectin involved in the mechanism of Alzheimer’s disease exacerbates AD pathology and impairment.

**Figure 3 ijms-21-06419-f003:**
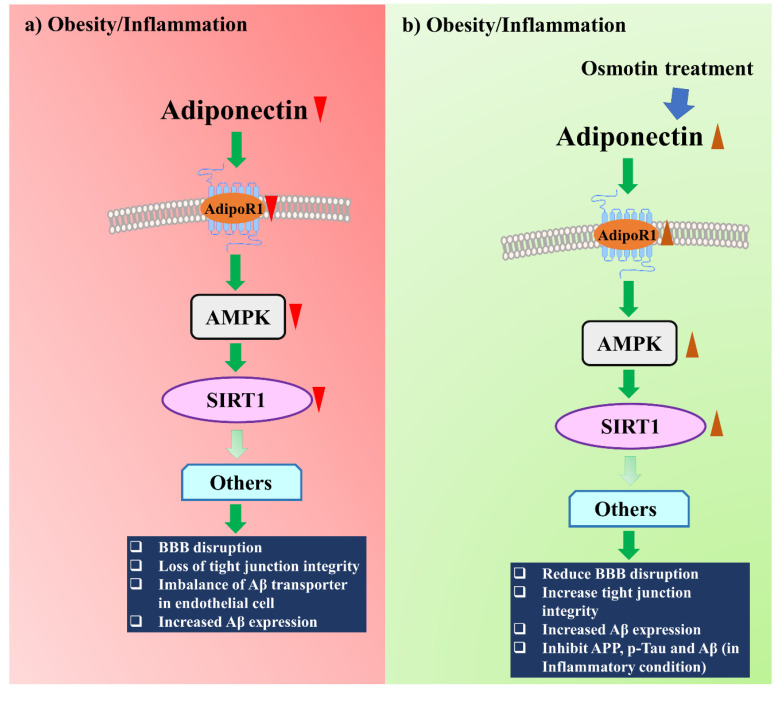
Osmotin treatment positively modulates AD through the AdipoR1/AMPK/SIRT1 pathway. (**a**) Obesity reduces adiponectin and AdipoR1 expressions, which negatively modulate the AMPK/SIRT1 pathway and increase the AD biomarkers. (**b**) Osmotin treatment increases adiponectin and AdipoR1 expressions and reduces AD biomarkers through the AMPK/SIRT1 pathway.

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
