# Peer review of "Adiponectin: The Potential Regulator and Therapeutic Target of Obesity and Alzheimer’s Disease"

_ijms, 2020, doi:10.3390/ijms21176419_

Round 1
Reviewer 1 Report
Dear Authors, I reviewed your interesting manuscript "Adiponectin: The potential regulator and therapeutic target of obesity and Alzheimer's disease".
Your paper raises some important issues in the field of development of AD-like neurodegenerative processes and their link with metabolic disturbances, particularly obesity, insuline resistance and diabetes.
I report my short revisions/suggestions below:
1) In the introduction, authors face with the issue of the link between obesity and AD; then, in the subsequent section (chapter 2), they introduce the concept of metabolic syndrome (metS). I suggest to prelude the possible contribution of metS as risk factor to AD development in the Introduction, when you deal with obesity and diabetes as risk factors (line 37, pag.1).
Instead, I suggest to further explain the link between metS and AD in the following section (pag3, chapter 2), also with a brief focus on the overlap that can be found between the clinical symptoms and imaging pictures in the two diseases (line 97-103).
Please consider the following citations:
Nasoohi S, Parveen K, Ishrat T. Metabolic Syndrome, Brain Insulin Resistance, and Alzheimer's Disease: Thioredoxin Interacting Protein (TXNIP) and Inflammasome as Core Amplifiers. J Alzheimers Dis. 2018;66(3):857-885. doi:10.3233/JAD-180735
Willette AA, Xu G, Johnson SC, et al. Insulin resistance, brain atrophy, and cognitive performance in late middle-aged adults. Diabetes Care. 2013;36(2):443-449. doi:10.2337/dc12-0922
de la Monte SM. Insulin Resistance and Neurodegeneration: Progress Towards the Development of New Therapeutics for Alzheimer's Disease. Drugs. 2017;77(1):47-65. doi:10.1007/s40265-016-0674-0
2) In the abstract, line 13: Pleas replace "tau" with "intracellular neurofibrillary tangles deposition". Line 14, please add "pathophysiological" before "development".
3) Pag. 2, line 48-49. Authors state that current disease modifying trials for AD are unsuccesful. However, some trials are still ongoing whereas other trials seem to have found promising results. So far, I suggest to change your statement by better specifying this issue.
4) pag.3, line 74: references seem to be not suitable for the sentence, since citations refer to the research diagnostic criteria for AD. Line 73, what do you mean with "special learning"? Please change (i.e., simply "learning"). The last sentence (line 115-116) expresses a general asertion, so it should be shifted at the beginning of the section.
5) Pag. 4, line 137: the sentence seems less clear, please reformulate (what do you mean with "memory of AD"?).
6) pag. 5-6, section 4.3. Please consider whether adiponectin might alter glucose metabolic patterns of brain FDG-PET scan. It could be an interesting issue.
7) pag. 9, line 297, pleas replace "mementine" with "memantine".
8) Please change the referencesas follows: pag. 2, line 62: [8-13], pag. 3, line 74-75: [14-17], line 109-110: [28-29], pag.4: [41-43], [44-46], [48-52], [54-57], pag. 5: [65-67], [75-78], pag.6: [100-103], [108,109], [110,111], pag.7: [113-115], [119,120], pag.9: [133,134], [141-143], [136-138], [146-147], [151, 153-156], [157-160], [161-163], [166-169], pag. 10: [177-179], [172-174], [180-182], [183,184], [185-188], [193-195], [196,197], [199-201], [202,203].
9) Please verify citation 13 and 116, they seem to be the same reference.
Author Response
Dear Reviwer
We thank the reviewers for fruitful advice, especially for suggesting better terms and sentences. We have revised the manuscript IJMS-855130 on the basis of the referees’ comments. Our specific responses are given below.
Hopefully, you will consider that our manuscript is now suitable for publication in the “International Journal of Molecular Science”.
Our responses to the referee's comments are submitted
Sincerely yours,

Reviewer 2 Report
A manuscript is a review focused on the potential importance of adiponectin in the context of Alzheimer's disease.
In general, the idea of the manuscript is interesting and important as metabolism is gaining increasing attention in the field of AD. Nevertheless, in my opinion the manuscript is not well organized and exstensive editing is required in order for the work to be easily readable and appropriately understood by the readers. There are duplicated sentences at several places throughout the manuscript (eg. [230]; [260] - here the authors even have two different references for the same sentence.;...)
Moreover, although the concept that adiponectin might be important in AD is attractive, at the current moment, there is very little direct evidence of this in the literature. The manuscript is not written in the style that reflects this objectively. In contrast, the authors write overly enthusiastic and make many questionable claims. This subjective approach is not appropriate and might encourage the readers to form erroneous conclusions.
[62] "...but the adiponectin might be a unique biomarker for AD." Why exactly?
[160] "In AD, Ab has to cross the BBB to be transported in the brain...". The following sentance explains that adiponectin protects the BBB. If something like this is written it has to be explained appropriately. This way it seems that it was only written to make the effect of importance of adiponectin greater than it really might be. Additionally, such serious statements [160] should have appropriate references.
[220] "Taken together, these data show that adiponectin is a key player in modulating the development of neurocognitive disorders and influencing their severity." No, the data you presented does not show that. The data suggests adiponectin might be involved in some way. However, whether it is just a confounder that loads together with metabolic phenotype, or if it is an etiopathogenetic factor is not at all clear in the current moment. This is a serious overstatement that might be taken for granted by others and do a lot of harm.
[230] Duplicated sentence
[240] Previous studies have attempted to use adiponectin level as an AD marker. -> Reference?
[260] Duplicated sentence again. Now with two different references.
[288] "This result can be interpreted as an adiponectin-related effect." No, it absolutely can not be interpreted this way.
Author Response
Dear Reviwer
We thank the reviewers for fruitful advice, especially for suggesting better terms and sentences. We have revised the manuscript IJMS-855130 on the basis of the referees’ comments. Our specific responses are given below.
Hopefully, you will consider that our manuscript is now suitable for publication in the “International Journal of Molecular Science”.
Our responses to the referee's comments was submited
Sincerely yours,
